# Synergistic Effect of Carbonate Apatite and Autogenous Bone on Osteogenesis

**DOI:** 10.3390/ma15228100

**Published:** 2022-11-16

**Authors:** Ikiru Atsuta, Tokihisa Mizokami, Yohei Jinno, Bin Ji, Tingyu Xie, Yasunori Ayukawa

**Affiliations:** 1Division of Advanced Dental Devices and Therapeutics, Faculty of Dental Science, Kyushu University, Fukuoka 8128582, Japan; 2Section of Implant and Rehabilitative Dentistry, Division of Oral Rehabilitation, Faculty of Dental Science, Kyushu University, Fukuoka 8128582, Japan; 3Mizokami Dental Office, Fukuoka 8190366, Japan

**Keywords:** autogenous bone, carbonate apatite, bone-inductive, bone formation, rat tibia

## Abstract

Bone augmentation using artificial bone is an important option in dental defect prostheses. A bone substitute using carbonate apatite (CO_3_Ap), an inorganic component of bone, was reported to have promising bone formation and bone replacement ability. However, the osteoinductivity of artificial bone is less than autogenous bone (AB). In this study, CO_3_Ap with AB is demonstrated as a clinically effective bone substitute. For in vitro experiments, an osteoclast-like cell (RAW-D) was cultured in the presence of AB, CO_3_Ap, or both (Mix), and the number of osteoclasts was evaluated. Osteoblasts were also cultured under the same conditions, and the number of adherent cells was evaluated. For in vivo experiments, a few holes were created in the rat tibia and AB, CO_3_Ap, or Mix were added. At 0, 14, and 21 days, the tissue morphology of the wound area was observed, and the thickness of the cortical bone was measured. In vitro, Mix did not increase the number of osteoclasts or osteoblasts. However, in vivo, the rate of bone replacement remarkably increased with Mix on dome-shape. A bone-grafting material combining osteoinductive AB with abundant artificial bone is expected to be clinically easy to use and able to form bone.

## 1. Introduction

Insufficient bone volume is a serious problem in dental prosthetic treatments. Jawbones resorb rapidly after teeth extraction [1,2]; however, it is important to recover or maintain the bone volume for denture stability, aesthetics of the crown prosthesis, and implant support. As a result, bone substitutes are used. Autogenous bone (AB) is the most widely used bone substitute [3]. However, because AB requires the patient’s own bone to be harvested, there are limitations because of invasiveness to the patient and how much can be harvested [4,5,6]. As a result, artificial bone is a promising alternative to maintain space until the bone is reformed [7,8]. Artificial bone is simply a scaffold for osteoblasts to seed and form bone, i.e., osteoconductivity [9], and artificial bone is slowly replaced by bone as a function of its bone replacement ability [10].

Osteoinduction is a process that supports the body’s healing process and acts as a starting point for bone formation [11,12]. Osteoinductivity is the ability to attract mesenchymal stem cells to their surroundings, differentiate the mesenchymal stem cells into osteoblasts, and induce osteogenesis [13]. This behavior is strongly observed in AB with bone marrow [11,14], and rarely observed without bone marrow or with artificial bones [15]. If artificial bone can be endowed with osteoinductivity, the efficiency of bone formation will dramatically increase. Specifically, artificial bone physically inhibits the invasion of soft tissue until new bone is formed, serves as a scaffold for proliferating bone-related cells, and may be an optimal bone-grafting material that can induce bone.

In this study, carbonate apatite (CO_3_Ap) was selected as a representative of artificial bone. Some materials for artificial bone remain at the site without being replaced for a long period of time to keep “space making” for bone formation, but there is a risk of infection [16,17]. It is important to have a composition similar to that of bone for biocompatibility. Therefore, an artificial material with a composition similar to that of autogenous bone, such as a bone substitute containing CO_3_Ap, which is an inorganic component of bone, has gained clinical attention [18,19]. CO_3_Ap is an inorganic component of bone with high osteoconductivity and bone replacement ability. However, CO_3_Ap is not osteoinductive like most other bone substitutes [20].

By blending CO_3_Ap and AB, CO_3_Ap will create the space for bone formation, and the AB will attract mesenchymal stem cells and induce bone formation by differentiating the cells into osteoblasts. Therefore, the advantages of each material are used.

The effect of mixing AB and artificial bone was evaluated using a bone defect in a rat tibia, and the mixture is expected to be clinically easy to use and efficacious in creating new bone.

## 2. Materials and Methods

### 2.1. Materials

AB was collected from the femur and tibia of 6-week-old Wistar male rats (different animals than used for the surgery experiments) using a bone scraper (Micross, OKABE, Tokyo, Japan). The collected AB was dried under UV irradiation on a clean bench for two days. Cytrans^®^ (CO_3_Ap, GC, Tokyo, Japan), a pure CO_3_Ap dense granule, was used as the bone substitute [21,22]. The granules were 300–600 μm in diameter.

### 2.2. Osteoblast and RAW-D Culture

MC3T3-E1 cells were used as an osteoblastic cell line, and they were cultured in α-modified minimum essential medium (αMEM; Wako, Osaka, Japan) with 10% fetal bovine serum (FBS, Thermo Fisher Scientific, Waltham, MA, USA) for four days.

RAW-D cells, an osteoclast precursor cell line, were differentiated to murine osteoclasts as previously described [23,24]. Briefly, RAW-D cells were cultured in α-MEM with 10% FBS in the presence of RANKL (50 ng/mL, ORIENTAL YEAST, Tokyo, Japan) for four days.

The cells (6.8 × 103/mL) were cultured indirectly with materials using a Transwell^®^ insert as a separator [25]. The cells were cultured in the bottom chamber with or without materials (AB, CO_3_AP, and AB/CO_3_AP) in the upper chamber (a pore size of 200 μm).

### 2.3. Immunofluorescence Staining

The Cell membrane and Actin filament of the osteoblasts were stained with CellLight Plasma membrane-GFP (Thermo Fisher Scientific) and tetramethylrhodamine isothiocyanate ghost pencil peptide (Chemicon International, Temecula, CA, USA). After staining, the nuclei were stained with VECTASHIELD^®^ (Vector Laboratories, Bulingame, CA, USA). The cells stained were observed under fluorescence microscope (BZ-9000, Keyence, Osaka, Japan).

### 2.4. Tartrate-Resistant acid Phosphatase (TRAP) Activity

The TRAP activity of the RAW-D cells was examined [26]. After four days of incubation, the cells were fixed in 37% formaldehyde (WAKO), permeabilized in ethanol/acetone (WAKO), and observed with a TRAP staining kit (Sigma-Aldrich, St. Louis, MO, USA).

### 2.5. Cell Counting

The samples were observed with BZ-X800 (KEYENCE). The number of TRAP-positive cells and osteoblast-like cells was counted per well with BZ-X800 Analyzer software (KEYENCE).

### 2.6. Animals

Wistar rats were cared for under the guidelines of Kyushu University (approval number: A29-222-0). The experimental model followed a previous report [27]. Briefly, bone defects (1.5-mm hole) were created in the right tibia of 6-week-old male rats (30; 120–150 g) using a dental round bur, and AB, CO_3_Ap, or their mixture (Mix) were used to fill the defects with 1.2 mm diameter NIET carrier (Japan dental supply, Tokyo, Japan). The carrier was used twice to keep the volume of the material in the defect constant. Bone-grafting material was added to form a dome over the defect, and healing was observed after fixation with a membrane (Cytrans Elashield; GC) and two sutures (4-0 Softretch: GC). The control group was allowed to heal without additional material. The rats were sacrificed via anesthesia after 14 or 21 days, and samples were created. Samples from animals sacrificed immediately after making the holes were considered Day 0.

### 2.7. Tissue Preparation

Tissues for hematoxylin and eosin (HE) staining were prepared as previously described [28,29,30]. Briefly, the samples were removed from the rats, fixed in 4% paraformaldehyde for 24 h, and then preserved in Kalkitox (FUJIFILM Wako Pure Chemical, Osaka, Japan) at 4 °C for 18 h. Sagittal sections (10 μm thick) were cut using a cryostat and stained with HE.

### 2.8. Bone Thickness

The vertical length of cortical bone on the bone marrow was considered the bone thickness. The anteroposterior position for the measurement was located by the central part of the hole and it was measured using a BZ-X800 microscope (KEYENCE).

### 2.9. Statistical Analysis

All measurements were performed 3 times by 3 people, and the average values were graphed. Data are expressed as means ± standard deviation (SD). One-way analysis of variance with Scheffe’s post hoc was performed. *p* < 0.05 values were considered significant.

## 3. Results

### 3.1. Effects of the Bone Substitute on Osteogenic Cells

As shown in Figure 1, a cell culture experiment was performed to confirm whether the proliferation of osteoblasts and the ability to differentiate into osteoclasts were affected by the bone substitute. In addition to the three groups of bone substitute (bone alone (AB group), carbonate apatite alone (CO_3_Ap group), and a mixture of AB and CO_3_Ap (Mix group)), a control was prepared without material (Cont group) and cultured for four days. The CO_3_Ap group showed less osteoblast proliferation than the control group and Mix group, and less osteoclast differentiation than the AB group.

### 3.2. Effects of the Bone Substitute on Bone Healing

Using the animal experimental model shown in Figure 2, morphological images were observed after 14 and 21 days, as shown in Figure 3. Figure 3b shows an image in which the cortical bone was perforated with a round bar. A section was prepared parallel to the long axis, and the cortical bone was completely lost at a width of 1.0 mm. The bone marrow was penetrated. As shown in Figure 3c, bone closure occurred even in the control at Day 14; although there were traces of the defect, the closure was complete at Day 21. In the AB group, after 14 days, the AB remained, similar to a foreign substance; however, after 21 days, the cortical bone was thicker than that in the control group. In the CO_3_Ap group, the material clearly remained even after 21 days; however, the surrounding area was covered with mature bone. In the Mix group, material remained as in the CO_3_Ap group; however, thick bone incorporating the AB was formed.

Figure 4 shows the thickness of the bone. After 14 days, the thickness was maintained only in the Mix group. After 21 days, the thickness in the control group was significantly less than that in the CO_3_Ap and Mix groups.

## 4. Discussion

When bone has a defect because of trauma, such as a fracture, the edge of the bone facing the defect is destroyed by osteoclasts, and bone formation-related cells, such as osteoblasts, migrate into the defect [31,32]. The defect is then filled with a collagen-like structure, and the bone is closed by inorganic material deposition [33,34]. However, because bone closure is slower than that of soft tissue growth, in many cases, soft tissue enters or presses the space where the bone should heal, thereby blocking bone healing [35,36]. Therefore, the bone substitute serves as a framework to secure a space for bone formation, and functions as a scaffold for osteogenic-related cells that migrate to the defect site to differentiate and secrete collagen. Additional important functions of material are an absorption speed to maintain space until bone formation progresses [7], cell affinity [37], and absorbability to allow for bone replacement [38].

Osteoinductivity to support osteogenesis is also an important material characteristic. The progression of bone formation from the site contacting the bone is general healing; however, with time, resorption of distant sites will occur. Therefore, it may be difficult to create bone with a significate thickness. Alternatively, if the material has osteoinductivity, bone formation will occur at the part facing the bone and from where the material is placed. Therefore, bone replacement will proceed faster and more efficiently, potentially reducing the risk that soft tissue invades the area.

However, AB with bone marrow is an outstanding material with osteoinductivity [39,40]. Although dried bone is clearly inferior to freshly harvested AB, it still has osteoinductivity [41,42].

As shown in Figure 2, a hole was created in the tibia, bone-grafting material was added to form a dome over the hole, and healing was observed after fixation with a membrane. Although a larger bone defect is necessary, the defect size was similar to that in other reports [11,39,43]. The healing period was approximately 14 days, even in the control group without bone substitute, and bone closure was observed in all experimental groups. However, as shown in Figure 3, the quality and volume of bone differed greatly. In the AB group, mature bone similar to that in the control group was formed after 14 and 21 days (Figure 3c). Alternatively, as shown in Figure 4, the thickness of the bone in the groups with CO_3_Ap was greater than that in the control group. A large volume of bone was formed because of the bulkiness of the material, and the cortical bone seemed to be thicker. After 21 days, there was a significant difference compared with the control group (Figure 4b). Moreover, in the Mix group with CO_3_Ap and AB, bone formation occurred over a large area that spread vertically, indicating that space-making and bone formation occurred at the same time. As shown in Figure 4a,b, a strong increase in vertical bone was observed in the Mix group, indicating optimal osteoinduction near the space-making artificial material.

However, it is unlikely that the material leached out and induced osteogenesis. As shown in Figure 1, when bones and materials were cultured at the same time using a Transwell without direct contact, the presence of the bone substitute did not significantly increase osteoblast proliferation or osteoclast differentiation compared with the control group. With CO_3_Ap, the proliferation of osteoblasts was significantly less than that of the control group (Figure 1a), and the proliferation of osteoclasts was less than that of the AB group (Figure 1b). CO_3_Ap elutes calcium in vivo [14,16]. Alternatively, the osteoblast proliferation and osteoclast differentiation increased with an increase in calcium concentration [44,45]. However, CO_3_Ap actually adsorbs calcium from a solution with high calcium concentration [46]. Specifically, in our recent study, we measured the calcium concentration in the culture medium, and found that the concentration decreased (data not shown).

There is a difference between cell culture conditions and an in vivo environment; however, rather than the material itself promoting osteogenesis, AB induced osteogenesis-related cells around CO_3_Ap. Future studies will clarify which factors induce such cells around the material, and they will be added to develop a novel bone substitute with further improved osteoinductivity.

## 5. Conclusions

A mixture of CO_3_Ap and AB takes advantage of the osteogenic and osteoinductive properties of each component, which may create reliable bone formation over a wide area.

## Figures and Tables

**Figure 1 materials-15-08100-f001:**
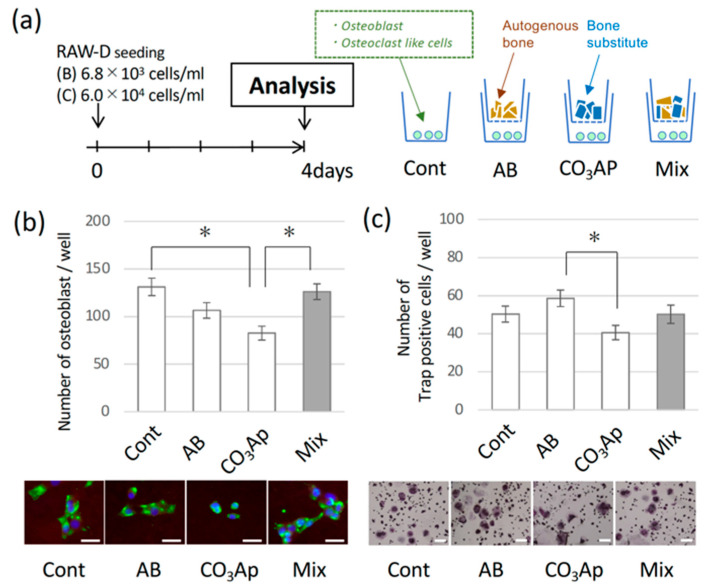
(**a**) Culture protocol. Osteoblasts from the rat bone marrow of a tibia were used, and RAW-D cells were used as osteoclast precursor cells. Autogenous bone (AB), carbonate apatite (CO_3_AP), and a mixture of AB and CO_3_AP (Mix) were placed in the upper well of a Transwell, and nothing was added to the control group (Cont). (**b**) Number of adhered osteoblasts after four days. Scale bar = 20 μm. (**c**) The number of TRAP-positive cells representing cells differentiated into osteoclasts. (* *p* < 0.05) Scale bar = 20 μm.

**Figure 2 materials-15-08100-f002:**
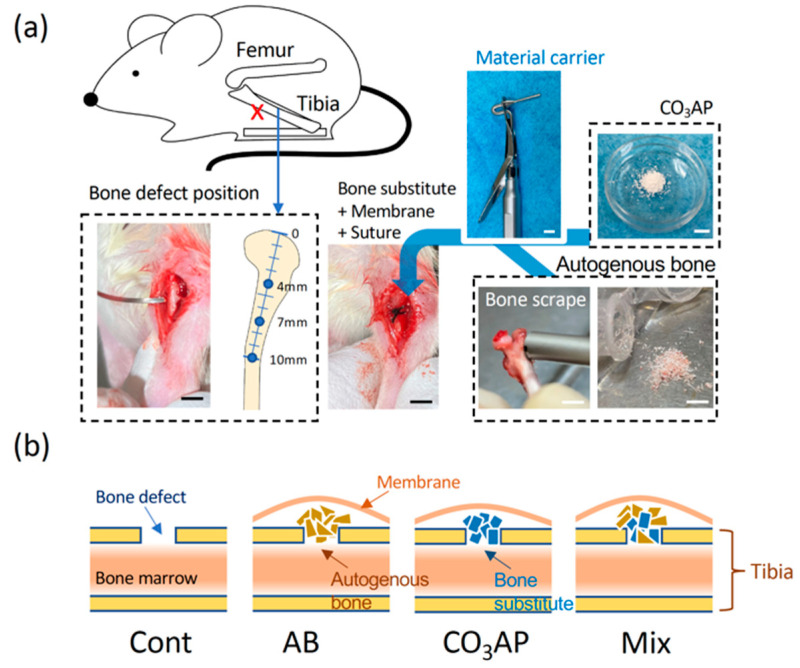
(**a**) Tibia defect and preparation of autogenous bone. A hole was created in the tibia (red X) using a round bur, material was inserted using a filling tool (material carrier), and the hole was covered with a membrane and two sutures. Scale bar = 5 mm. (**b**) Experimental group. For the control, nothing was inserted into the tibia hole. In the autologous group, a separate dry tibia was used. In the Mix group, a 1:1 volume of scraped bone and CO_3_AP was used to fill the hole.

**Figure 3 materials-15-08100-f003:**
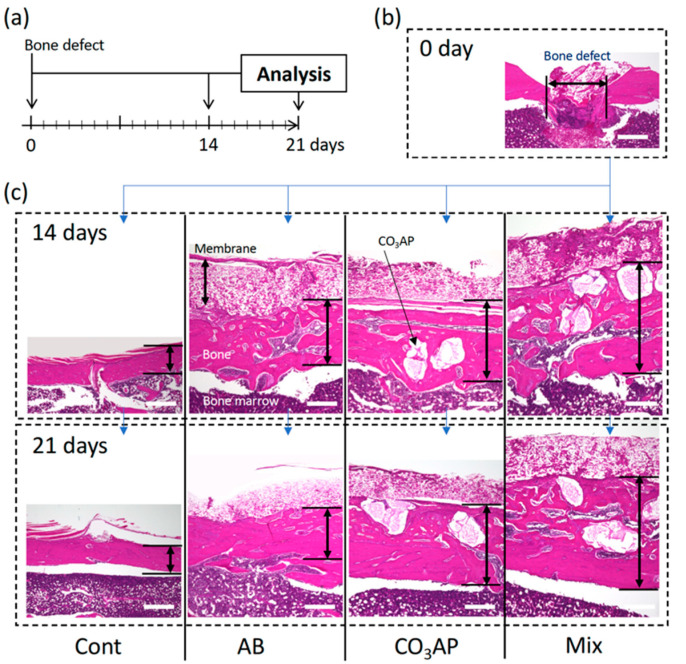
(**a**) Animal protocol. (**b**) Photograph of the morphology immediately after creating a bone defect in the tibia (Day 0). (**c**) Osteogenic process. Hematoxylin and eosin staining after 14 days and 21 days in the control group (Cont), autogenous bone group (AB), carbonate apatite group (CO_3_AP), and AB and CO_3_AP mixed group (Mix). Scale bar = 100 µm.

**Figure 4 materials-15-08100-f004:**
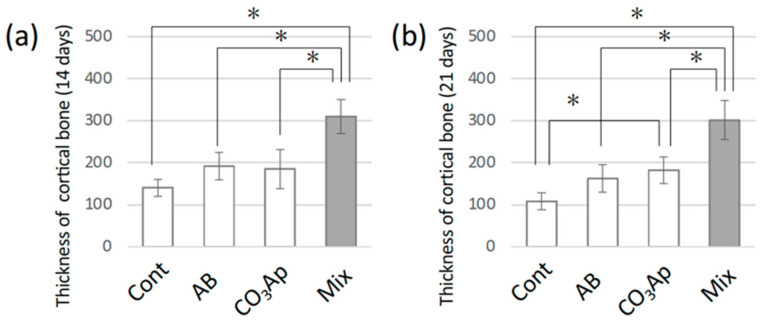
Thickness of the injured bone (**a**) 14 days and (**b**) 21 days after surgery (* *p* < 0.05) for the control group (Cont), autogenous bone group (AB), carbonate apatite group (CO_3_AP), and AB and CO_3_AP mixed group (Mix).

## Data Availability

The data presented in this study are available on request from the corresponding author.

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
