# Peer review of "Synergistic Effect of Carbonate Apatite and Autogenous Bone on Osteogenesis"

_materials, 2022, doi:10.3390/ma15228100_

Round 1

Reviewer 1 Report

In this manuscript was presented the role of carbonate apatite in bone regeneration. Also the authors compare this materials to autogenous bone graft or both. For a better confirmation of the result please add/explain the following:

- add more information in introduction  regarding the benefits of carbonate apatite, there are studies that recommend this material?

-add some imagine with osteoblast and osteoclast in the culture

- add Trichromic Massson staining to determine levels of osteoclasts

- why the transwell experiment was done in this way?

- how you count the number of osteoblast?

- please check the academic language: " immature mesenchymal stem cells", I suggest to remove "immature"

Author Response

Dear Reviewer #1

Thank you for your positive feedback and pointing out the flaws of our article. We had modified the manuscript according to your comment. These details included:

Comments to the Author

In this manuscript was presented the role of carbonate apatite in bone regeneration. Also the authors compare this materials to autogenous bone graft or both. For a better confirmation of the result please add/explain the following:

1)  - add more information in introduction regarding the benefits of carbonate apatite, there are studies that recommend this material?

Response: Thank you for pointing out. We have added more information of carbonate apatite in “Introduction” as follows:

“Some materials for artificial bone remain at the site without being replaced for a long period of time to keep “space making” for bone formation, but there is a risk of infection [16, 17]. It is important to have a composition similar to that of bone for biocompatibility. Therefore, an artificial material with a composition similar to that of autogenous bone, such as a bone substitute containing CO3Ap that is an inorganic component of bone, has gained clinical attention [18, 19].”

2)  -add some imagine with osteoblast and osteoclast in the culture

> Response: Thank you for your advice. We have changed the figure as suggested. And the method for fluorescent staining of osteoblasts is also described in the Materials & Methods section.

3)  - add Trichromic Massson staining to determine levels of osteoclasts

> Response: Thank you for your comment. In this study, osteoclasts were evaluated only in culture experiments using cell lines. Therefore, trap staining was used to determine osteoclasts. We understand the effectiveness of “Trichromic Massson staining”, so when we conduct animal experiments in the future, we will use that staining method as you indicated.

4)  - why the transwell experiment was done in this way?

> Response: Thank you for your comment. Since the materials used this time was in the form of a powder, culturing them together with cells made it impossible to measure and evaluate the cell behavior. between the part in contact with the material and the part not in contact, variation in the results of cell reactions. And the results are different between cells that are in contact with the material and cells that are not. Therefore, indirect culture using Transwell was performed in this time.

5)  - how you count the number of osteoblast?

> Response: Thank you for your comment. We have added small information of the cell-counting method as follows: “2.4. Cell counting. The samples were observed with BZ-X800 (KEYENCE, Osaka, Japan). The number of TRAP-positive cells and osteoblast like cells was calculated using BZ-X800 Analyzer software (KEYENCE).” By the way, unlike osteoclasts, the differentiation of osteoblast cannot be evaluated, so the number of cells was simply counted with a cell counter 3 days after seeding.

6)  - please check the academic language: " immature mesenchymal stem cells", I suggest to remove "immature"

> Response: Thank you for pointing this out. Both have been deleted.

Reviewer 2 Report

The study is integral to the scientific world, it has a scope and of benefits. I congratulate the author's team for this great effort. I would like to mention a few points of improvement further.

Title:

this paper is encompassing a trial, please follow the CONSORT guidelines for "title" writing.

methods:

as per CONSORT guidelines, please add a "flow diagram" of the study.

please check line number 1o6 for correction.

the description of the methodology is lacking the details of measuring bone thickness.

moreover, the calibration of the microscope, how it was performed?

I would also like to request the author's team to add, who performed the data collection amongst the team. additionally, how the examiner's calibration or reliability was performed, since the outcome of the study is based on observation/measurement of bone.

please check the paper for typo errors and English corrections. 

Author Response

Dear Reviewer #2

Thank you for your positive feedback and pointing out the flaws of our article. We had modified the manuscript according to your comment. These details included:

Comments to the Author

The study is integral to the scientific world, it has a scope and of benefits. I congratulate the author's team for this great effort. I would like to mention a few points of improvement further.

1)  Title: this paper is encompassing a trial, please follow the CONSORT guidelines for "title" writing.

2)  methods: as per CONSORT guidelines, please add a "flow diagram" of the study. please check line number 1o6 for correction.

> 1) & 2) Response: Thank you for your advice. That's a very important point for us. Receiving your suggestion, we checked some guidelines, but it seems that CONSORT does not apply to our basic research. In this time, our research follows the ARRIVE guidelines.

3)  the description of the methodology is lacking the details of measuring bone thickness.

> Response: Thank you for your comment. We have added small information of measuring bone thickness in the Materials & Methods section.

4)  moreover, the calibration of the microscope, how it was performed?

> Response: Thank you for your pointing out. In our study, all measurements were performed 3 times by 3 people with BZ-X800 Analyzer software (KEYENCE), and the average values were graphed. This measurement method has been added to the Materials & Methods section.

5)  I would also like to request the author's team to add, who performed the data collection amongst the team. additionally, how the examiner's calibration or reliability was performed, since the outcome of the study is based on observation/measurement of bone.

> Response: Thank you for your advice. At the time of the first submission, the experimenter alone took pictures for measurement. Therefore, we re-measured all the data in this re-submitted paper. The details of the measurement method are as described above. The details of the data changed, but the results of the study remained the same. The diagram has also changed slightly.

6)  please check the paper for typo errors and English corrections. 

> Response: The text was proofread by native English speakers. The manuscript has been proofread.

Round 2

Reviewer 1 Report

The manuscript was impoved after adding the new comments and results, and I recommend for publication in this form.